# Performance Comparison of Directed Acyclic Graph-Based Distributed Ledgers and Blockchain Platforms

**Felix Kahmann** *[ID], **Fabian Honecker** [ID], **Julian Dreyer** *[ID], **Marten Fischer** *[ID] and **Ralf Tönjes** *[ID]

Faculty for Engineering and Computer Sciences, University of Applied Sciences, 49076 Osnabrueck, Germany; fabian.honecker@hs-osnabrueck.de

* Correspondence: felix.kahmann@hs-osnabrueck.de (F.K.); j.dreyer@hs-osnabrueck.de (J.D.); m.fischer@hs-osnabrueck.de (M.F.); r.toenjes@hs-osnabrueck.de (R.T.)

**Abstract:** Since the introduction of the first cryptocurrency, Bitcoin, in 2008, the gain in popularity of distributed ledger technologies (DLTs) has led to an increasing demand and, consequently, a larger number of network participants in general. Scaling blockchain-based solutions to cope with several thousand transactions per second or with a growing number of nodes has always been a desirable goal for most developers. Enabling these performance metrics can lead to further acceptance of DLTs and even faster systems in general. With the introduction of directed acyclic graphs (DAGs) as the underlying data structure to store the transactions within the distributed ledger, major performance gains have been achieved. In this article, we review the most prominent directed acyclic graph platforms and evaluate their key performance indicators in terms of transaction throughput and network latency. The evaluation aims to show whether the theoretically improved scalability of DAGs also applies in practice. For this, we set up multiple test networks for each DAG and blockchain framework and conducted broad performance measurements to have a mutual basis for comparison between the different solutions. Using the transactions per second numbers of each technology, we created a side-by-side evaluation that allows for a direct scalability estimation of the systems. Our findings support the fact that, due to their internal, more parallelly oriented data structure, DAG-based solutions offer significantly higher transaction throughput in comparison to blockchain-based platforms. Although, due to their relatively early maturity state, fully DAG-based platforms need to further evolve in their feature set to reach the same level of programmability and spread as modern blockchain platforms. With our findings at hand, developers of modern digital storage systems are able to reasonably determine whether to use a DAG-based distributed ledger technology solution in their production environment, i.e., replacing a database system with a DAG platform. Furthermore, we provide two real-world application scenarios, one being smart grid communication and the other originating from trusted supply chain management, that benefit from the introduction of DAG-based technologies.

**Keywords:** directed acyclic graphs; IOTA; blockchain; Ethereum; Hyperledger Fabric; performance; throughput; latency; distributed ledger

## 1. Introduction

Distributed ledger technologies have evolved in many different directions since the introduction of the Bitcoin blockchain in 2009. The most common new blockchain ecosystems, such as the Hyperledger project suite or various Ethereum-based blockchain systems, are considered the next evolution of distributed ledger technology (DLT) in general. However, with all of these technologies relying on the same fundamental idea of chaining multiple blocks containing transactions in chronological order, each of these technologies faces the same scalability problems.

With an increasing demand for new DLT platforms, the networks need to cope with a growing number of network participants as well. Scaling the networks to arbitrary numbers

of participants is a well-known problem with one-dimensional blockchains. Prominently, the effects can be seen in the Ethereum network, which has experienced increased popularity in the last decade [1]. By introducing new consensus mechanisms, e.g., proof-of-stake (PoS) instead of proof-of-work (PoW), modern blockchain platforms try to remediate the scaling effects and achieve a generally higher transaction throughput. However, these changes do not resolve the scalability issue in the long term [2,3].

The fundamental problem remains the same for all blockchain platforms: the linear, non-parallel underlying data structure, which cannot be trivially parallelized. Therefore, new concepts involving a more liberal form of data structure, called directed acyclic graphs (DAGs), form the basis of potentially more performant DLTs, which can be scaled beyond the limits of current blockchain-based systems while also allowing a higher throughput of transactions per second. Using a DAG that allows for more than one edge per vertex enables full parallelization and, thus, significantly better scalability of the system in theory.

One of such technologies is IOTA, which employs a pure DAG as its main data structure. Also, hybrid approaches that keep compatibility with the Ethereum ecosystem by using a main blockchain for data storage but rely on a DAG for the consensus operation have also seen a gain in popularity. The most prominent examples of such technologies are Fantom and Avalanche, which are both publicly available. With this larger set of possible DLT variants, evaluating the scalability and performance of each system individually is a necessary task for any developer intending to select a suitable DLT for any given use case [4,5].

Therefore, the main research objective of this article is to find concrete, real-world applicable performance numbers for DAG-based and hybrid DLTs that can be compared to the performance metrics of blockchain platforms. By evaluating the numbers using common metrics, such as the transactions per second (TPS), real-world performance numbers will result, most notably for DAGs, which can be compared to the theoretically expected numbers.

In order to quantify the real performance benefit of the new underlying data structure, this article aims to show a broad spectrum of performance evaluation numbers for each of the five evaluated technologies, either blockchain-based, DAG-based, or hybrid DLTs. By using the most prominent technologies in each category, in particular Hyperledger Fabric, Ethereum, IOTA, Fantom, and Avalanche, and evaluating the system in terms of throughput and latency, this article creates a universal basis for other performance evaluations to compare. Furthermore, we tested the three technologies in their respective private networks to evaluate the scaling effects of increasing the number of network participants. In the case of Ethereum, different consensus mechanisms were also used to identify potential bottlenecks created by the given algorithm in use. Finally, this article also provides two particular real-world use cases that can benefit from the introduction of DAG-based DLTs as their primary means to store data. The main contributions of this article can be summarized as follows:

- In-depth description of the various DLT data structure paradigms;
- Performance evaluation in terms of throughput (TPS and latency) of the introduced DLTs;
- Scalability evaluation of the private DLTs;
- Use-case description for supply chain management and smart grid communication application scenarios that benefit from the introduction of DAG-based DLTs.

The remainder of this article is structured as follows: Section 2 provides an in-depth research overview in the domain of Hyperledger Fabric (HLF) and Ethereum blockchains of DAG-based DLTs and their performance implications. After that, Section 3 describes all the necessary technical background for the conducted performance evaluation that is theoretically described in Section 4, as well as the used methodology. The concrete performance numbers are then shown and explained in Section 5, followed by two exemplary application scenarios in Section 6, which can benefit from the introduction of DLT in their concepts. Finally, Section 7 concludes this article.

## 2. Related Work

Popular blockchain systems and platforms, such as Bitcoin and Ethereum, are well known to have scalability issues. In their original variants, both systems used a PoW consensus algorithm that heavily influenced the performance of the technology [6,7]. For Bitcoin, the PoW allowed only seven TPS to be validated on average [3], whereas Ethereum was capable of handling up to thirty TPS [8]. For modern, global payment systems, those numbers will not meet the requirements for instant money transfers. Thus, current research proposes different means to remedy the scaling problem of the aforementioned technologies. In their review, Yang et al. [6] discussed several concepts for improving the scalability of Ethereum and Bitcoin. Most notably, DAG-based data structures allow a more resilient and less error-prone execution of transactions in a parallel manner. Thus, DAG-based DLT systems are considered to be more scalable than their blockchain counterparts. The authors of [6] also discussed different approaches for off-chain payment networks such as Lightning (for Bitcoin) or Raiden (for Ethereum) networks that both form a side-chain handling the monetary transactions. By using advanced blockchain up/downstream smart contracts, transaction times and costs are minimized to an acceptable and real-world usable level.

Additional research focused on integrating blockchain, or DLT in general, into modern digital systems. For example, in their work, Malik et al. [9] proposed to integrate blockchain-based smart contracts into smart grid applications. They argue that, by creating a decentralized energy market, a decentralized smart contract platform will enhance the overall redundancy and trustworthiness of the operations. In their proof-of-concept implementation, the authors utilized both HLF and Ethereum for smart contract execution. Due to the public network setup of Ethereum, the performance in terms of flexibility and transaction throughput was significantly lower (6 TPS) in direct comparison to HLF (96.7 TPS). However, the authors did not mention how to scale the fabric network to a public scale and thus enable a more decentralized network setup. Other work by Dabbagh [10] and Choi [11] also evaluated the raw performance numbers of the Ethereum blockchain both in public and private network setups. The findings of [10] suggest that HLF in versions 1.4 and below significantly outperforms public Ethereum transaction speeds. However, the comparison between both technologies is inherently unfair since Ethereum in its public variant involves a significantly larger number of consensus nodes and a completely different network setup in general. By comparing the raw TPS numbers of [11] of the private Ethereum network to the previous HLF TPS numbers of [10], a fair comparison can be made. The findings of [11] suggest that pure query operations can reach a TPS number of over 1000. It shall be noted that the hardware used in the test runs of Choi et al. was significantly more powerful than the hardware used by Dabbagh.

The performance of HLF ("Fabric" for short) in its various versions up to 2.2 was tested and evaluated in past research [12]. The findings suggest that, compared to other DLTs such as Ethereum, Fabric is far more reliable and performant in terms of TPS and latency [13], whereas Ethereum enables a significantly easier entry point and a higher degree of decentralization [14]. All conducted studies relied on different Linux-based hardware setups and also different HLF network setups. The latter has been shown to be a notable point to consider when approximating the required performance of the network. Given the application scenario, an HLF network can be configured to meet the desired needs in terms of transaction throughput or latency times. In the case of Industry 4.0 application scenarios, Dreyer et al. proposed a decision algorithm for determining the minimum network setup for a given scenario [15].

HLF is designed to be used in almost arbitrary use cases that require trusted or private data storage. Recent work by Alsallut et al. provided a comprehensive overview of use cases within food supply chain management that make use of Fabric. In their paper, the authors mention use cases of Walmat using HLF for food traceability or the Malaysian Halal industry using HLF to ensure the quality of the supplied food. In each case, the added trust provided by Fabric leads to a higher degree of confidence compared to traditional database systems.

More recent approaches use a directed acyclic graph (DAG) as a base structure for the DLT. Such approaches, like IOTA, for example, emerged to address the scalability problems of the blockchain data structure [16]. The work of Živić et al. evaluated such DAG-based DLT with regard to their applicability for IoT. The work concluded that a DAG outperforms the classical blockchain and is more suitable for IoT environments due to increased throughput while maintaining low transaction costs at the same time. They also outlined as the current development state that implementations like IOTA remain in an experimental state and do not provide full decentralization yet limit the current scalability. Decentralization was identified as a challenge for the upcoming years [17].

Park et al. designed a DAG-based DLT for use in smart grid systems to manage energy trade in the form of transactions. The work presented the so-called PowerGraph DLT, with a new consensus algorithm to reduce the validation delay of traditional systems, especially those designed for use in smart grid environments. The PowerGraph DLT was proven to have a higher transaction processing rate than other technologies [18].

Silvano et al. conducted a survey based on several research papers, identifying the areas of usage of IOTA as well as the advantages and disadvantages of its use. They identified the Internet of Things (IoT), machine-to-machine (M2M), and e-health as key fields of usage. The listed advantages included high transaction rates, feeless transactions, resource efficiency and security, and the ability to share data. The disadvantages include the missing decentralization, the absence of smart contracts, low LPWAN compatibility, and the missing reuse of transaction addresses [4]. In contrast, in this paper, we aim to provide a comparison of multiple popular DLTs in order to find scalability differences among them. In addition, we will focus on the PoW used in IOTA.

Wang et al. provided an evaluation of the scalability of IOTA by building a private network on real hardware and using different self-developed testing tools against it. Their findings include that IOTA provides a lower TPS than that provided by the whitepaper, archiving a throughput of around 15 using their experimental setup. Also, they identified the database queries used to check the uniqueness of an address as a main bottleneck [19]. In comparison to that, the current paper will directly compare IOTA to other DLTs with a predefined set of key performance indicators (KPIs) used for general DLT comparison.

Regarding further transactions on IOTA, Sarfraz et al. focus on the privacy of IOTA and improvements that could be made to it. They propose a protocol using a decentralized mixing approach in order to prevent identification while preserving decentralization [20].

## 3. DLT Implementations and Theoretical Performance

The current state of modern DLTs is under high development and, thus, is changing rapidly. The following subsection will, therefore, provide a mostly factual overview of the current development state for each of the DLTs evaluated in this article. The individual technologies can be separated into three distinct categories: DAG-based, blockchain-based, and hybrid technologies. These categories refer to the implementation of the ledger, which is required for the network-wide consensus of the data. The following Table 1 provides an overview of all evaluated technologies and their respective DLT categories.

**Table 1.** Evaluated DLT technologies and categories.

| DLT | Category | Version |
|---|---|---|
| Hyperledger Fabric | Blockchain | 2.3 |
| Ethereum | Blockchain | 23.1.1 |
| IOTA | DAG | Crysalis |
| Fantom | Hybrid DAG + Blockchain | Lachesis |
| Avalanche | Hybrid DAG + Blockchain | Snowball |

*3.1. Blockchain Platforms*

This article refers to modern "Blockchains", such as Ethereum, Polygon, or others, as blockchain platforms, since they allow custom-made smart contracts to be run on them. For the purpose of precision, the term blockchain is used to refer to the underlying data structure. As such, traditional Bitcoin would be considered to be a blockchain due to its lack of smart contract functionality, whereas Ethereum is characterized as a fully-featured blockchain platform.

### 3.1.1. Ethereum

Ethereum was one of the first blockchain technologies that brought notable change to the whole ecosystem. With the introduction of the Ethereum blockchain platform in 2015, Vitalik Buterin introduced a novel way of executing so-called "Smart Contracts" on a blockchain. At that time, this approach was a true revolution, since previously introduced blockchain technologies, such as Bitcoin, were mainly considered to be non-programmable or just seen as an advanced monetary data storage concept. With Ethereum, developers had the opportunity to write custom code and execute it on the Ethereum blockchain using the Ethereum virtual machine (EVM), a custom virtual execution environment running on every Ethereum Miner node, and Solidity, a custom programming language for the EVM.

In its initial version 1.0, Ethereum used a proof-of-work consensus scheme, enabling block validation times of twelve to fifteen seconds on average [21]. When considering the limited number of transactions that can be fit into one block, there is a theoretical maximum of 30 TPS. Whilst being significantly faster than Bitcoin in terms of block validation speed and TPS [22], Ethereum is considered to be one of the largest sources of wasted energy worldwide [23], due to its inefficient mining approach. To combat these concerns, Ethereum 2.0 was proposed, introducing a new proof-of-stake protocol called "Casper" [24]. Using the new consensus protocol in conjunction with a new blockchain sharding scheme, Ethereum is capable of executing more transactions with a significantly lower energy footprint. Sharding allows the distribution of parts of the blockchain to smaller shards, generally leading to better scalability of the system. However, as Yu et al. describe in their paper, the smaller shards on a given blockchain platform have a higher security risk compared to a non-sharded blockchain, most notably during cross-shard communication [25].

With Ethereum 2.0, new validator nodes stake their Ether tokens to finally be allowed to validate a new block and append it to one of the sharded blockchains on one of the other validator nodes. Ultimately, this scheme results in more efficient energy use in comparison to a PoW-based mining protocol since no computationally heavy task needs to be solved by an arbitrary number of miners anymore. Furthermore, sharding the blockchain enables higher validation and block finality times due to the parallelization of the mining process [26]. New Ethereum 2.0 clients were developed to implement the features in different execution environments. Previously conducted studies show that some client software is still in an early development state and requires some rework to achieve more reliable and resilient execution. However, significantly lower blockchain synchronization times have been measured that directly relate to the more scalable sharding concept [27].

The Ethereum network itself is set up homogeneously. Each participating node can act as a validator node or just listen for new blocks and execute smart contracts. Since the field of potential clients is vast, including low-resource smartphones issuing payments or large data centers validating the blockchain, different usage scopes need to be considered. In the previous Ethereum version 1.0, only the most recent part of the chain needed to be stored on the device, enabling a lower data footprint for low-resource devices. These are called "lite-nodes". To use this feature, so-called full nodes that host the complete blockchain and make it available for the lite-nodes are required. These can be hosted, e.g., in large data centers. This enables Ethereum to be publicly available and also feasible for resource-constrained devices.

In Ethereum, application programs (smart contracts) are written using the programming language Solidity and run on the Ethereum platform. Each smart contract has an individual *Gas* value that has to be provided by each caller of the given smart contract. Depending on the complexity of the smart contract itself, the Gas consumption to execute it may vary. Apart from the option to run an Ethereum smart contract on the public mainnet, a developer may choose to test or even deploy it on a local private Ethereum test network. Since Ethereum requires a minimum amount of *Gas* for each transaction, real-world money is required to exchange it for the required amount of Ether tokens. This would, in turn. render proper software testing of the smart contract ecologically unattractive. Therefore, using the essentially free Ethereum testnet is a better way to first test the smart contract and later deploy it on the mainnet.

The Hyperledger Foundation also introduced a private Ethereum client called *Besu* [28]. Besu can be used to set up custom Ethereum private networks for various use cases. By enabling a developer to set individual network parameters, such as the block size, block timeout, number of network participants, and even the consensus mechanism, Besu allows a high degree of customizability. Furthermore, the developer has full control over the issued tokens within the network and can intervene in any problems occurring at runtime.

3.1.2. Hyperledger Fabric

Hyperledger Fabric (Fabric for short) is one of many blockchain projects of the Hyperledger Foundation [29] and offers a fully customizable business blockchain platform. As one of many key concepts, Fabric also allows developers to run custom-made smart contracts, called Chaincode, written in modern programming languages such as Java, JavaScript, and Golang [30]. By using these general-purpose programming languages, well-established software libraries, e.g., for cryptographic algorithms, data handling, and arithmetic, can be used. The use of reviewed security libraries is especially advantageous.

Fabric leverages a custom *world-state* paradigm to store the data on the network. The world-state is a traditional key-value database that is used to store the real data sent during a transaction. Every data operation like `Create`, `Read`, `Update`, or `Delete` is logged by the underlying transactional blockchain, thus allowing a fully transparent and tamperproof versioning history. However, while traditional blockchain data structures allow only `Create` and `Read` operations, the world-state paradigm also allows arbitrary modifications of the data after initial insertion as well as deletion of data.

Fabric also uses a private/permissioned network architecture that hosts heterogeneous network participants, each with different roles. Generally, each Fabric network consists of at least one *Channel*. A Channel is used to host its own blockchain instance and must be joined by one or more *Peers*. These are the main network participants maintaining the blockchain's integrity. Formally, they can be divided into *Endorsement* and regular Peers. The Endorsement Peers are the network participants that execute and validate the desired chaincode, whereas regular Peers just keep a local copy of the ledger without executing any chaincode. Consensus in Fabric is established through the use of designated *Orderer* nodes that execute every consensus-related aspect. Their main purpose is to ensure that the underlyingpractical Byzantine fault tolerance (PBFT) algorithm finalizes and that the correct order of transactions is propagated to every Peer in a given Channel. For further organizational logic, Fabric also provides means to establish so-called *Organizations*. Each Organization hosts at least one Anchor Peer, which interacts with the Orderer. An Organization can be used to specify particular access right permissions (authenticated through a certification authority (CA)), install a specific chaincode, or improve network performance [12]. A conceptual overview of the Fabric network structure is provided in Figure 1.

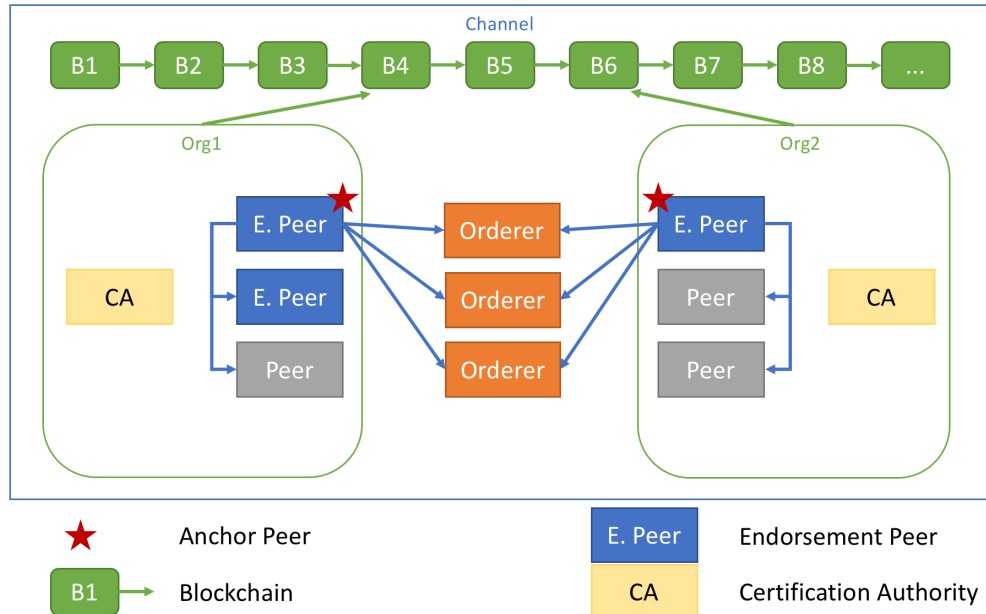

**Figure 1.** Overview of Fabric's organizational network structure.

In direct comparison to other blockchain platforms, Fabric does not require a native token to reach consensus among the different network participants but rather relies on implicit trust between the different peers [11]. Since the network access is permissioned and secured using a public–key infrastructure (PKI), no external, unauthorized party can join the network and potentially manipulate the blockchain [12]. Therefore, different consensus mechanisms can be employed in a private/permissioned network scenario than on public blockchain platforms, making them significantly more performant [10]. However, since network access is inherently restricted and thus less publicly available, decentralization of the network is a topic of concern. When hosted, e.g., in a centralized data center, Fabric will not be able to cope with any outages and may not recover from any local failure of the network.

*3.2. Directed Acyclic Graph-Based Platforms*

Compared to the well-known blockchain systems, a DAG is a different data structure for building a DLT system. The general idea is to overcome the aspect of having one entry or block after another linearly, like a blockchain [31]. This enables the possibility of having multiple predecessors for one single block and, thus, allowing multiple parallel appending operations at a time. By adjusting the amount of allowed predecessors of one block, the different new blocks can be validated and appended in parallel rather than within a single thread [32]. One example of such a DAG-based DLT is IOTA, which uses a DAG called the tangle. An overview of the DAG structure is given by Figure 2 [5].

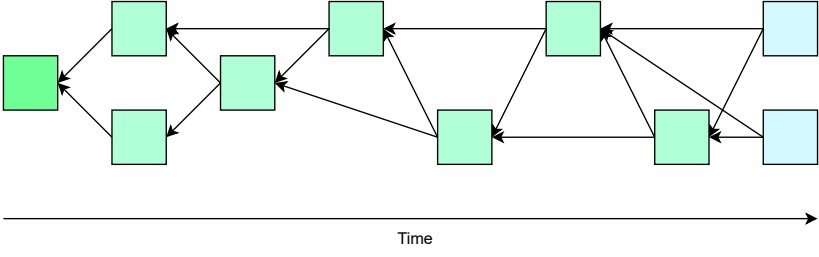

**Figure 2.** IOTA tangle with tips (blue), validated transactions (green), and the genesis at the front (leftmost, bolder green)

A new, not yet validated transaction in the tangle, marked as a blue block in Figure 2, is called a tip. The green blocks indicate transactions that have been validated already. The first transaction of the tangle is called genesis [33] (cf. the left-most block in Figure 2). In order to be appended and, thus, be validated, each tip must validate two previously created transactions. These validations are represented as the edges in the tangle. IOTA features different algorithms and strategies to select the candidates for validation from the set of all not-yet validated transactions [16]. A transaction can be considered valid if it includes references to previously validated transactions and a valid nonce value for the transaction hash [34]. A full validation criteria list and an overview of the detailed message structure can be found in [35].

The consensus mechanism used by IOTA can be classified as aproof-of-authority (PoA), which uses the identity of a node as stake [36]. IOTA uses a centralized node, called the coordinator, to constantly create special transactions called milestones. All other transactions in the tangle that are directly or indirectly referenced by such a milestone are considered valid in the network. The identity of the coordinator is known to all other participants in the network [37]. A notable point is that the utilization of a coordinator is considered a temporary solution and will be removed in the IOTA 2.0 update in the future and replaced by a PoS algorithm [38]. Additionally, IOTA uses a classical PoW to prevent the flooding of the network. The difficulty of the algorithm can be modified using the PoW score parameter. It defines the average amount of hash operations needed per byte to find a valid nonce value [39]. IOTA's main network is configured to have a default value of 4000, leading to an average of 4000 hash operations per byte. This implies a correlation between the length of a transaction payload and the needed amount of hash operations [40,41].

IOTA provides a development library available in different programming languages, including Python and Rust, which can be used to interact with the IOTA nodes [42]. The necessary software to run and host a custom node is provided by the IOTA Foundation, named Hornet [43]. To interact with the custom nodes, the iota.rs library is used in this article [44,45]. The Hornet node also offers the ability to run the PoW instead of the client if the node is configured to do so [46].

*3.3. Hybrid Distributed Ledger Architectures*

The implementation of smart contract capabilities in DLT requires strict consistency and order of transactions within the system. It ensures that all nodes maintain the same decisions for a certain position in the ledger, thereby increasing the trustworthiness and security of the system and enabling the conditions of the smart contract to be correctly assessed and executed. In a blockchain-based DLT, the order of transactions is implicitly given, as every transaction is carried out in a strict chronological order and each copy of the ledger on every node in the network is identical. However, DAG-based DLTs introduce some challenges regarding smart contract capabilities. Since transactions in DAGs do not necessarily take place in a synchronized order on every node, this can complicate the execution of smart contracts, which require strict consistency and arrangement. For this reason, a hybrid architecture is typically proposed, combining DAGs and blockchains [47].

The combination of DAGs and blockchains in hybrid DLTs represents a promising approach to improving scalability and throughput while ensuring proper smart contract execution. In these hybrid DLTs, DAGs are used to accelerate the consensus mechanisms, enabling higher scalability and throughput. The individual transactions are initially mapped on the DAG and consecutively arranged in strict order on a blockchain to ensure the necessary transaction order consistency for smart contract execution. In other words, while the DAG enables consensus on the state of transactions in the system, the blockchain ensures that this consensus is recorded in the strictly ordered manner required for the execution of smart contracts. In the following, we present two DLT protocols, Fantom and Avalanche, that use this hybrid approach [47].

### 3.3.1. Fantom

Fantom is a high-performance, scalable, and secure DLT platform that is built on the "Lachesis" consensus algorithm and a hybrid DLT architecture using an event-based directed acyclic graph (EventDAG) [48,49]. An EventDAG is a structure in which different nodes (referred to as events or event blocks in this context) are connected by edges that point to previous parent events. Each node represents a consensus message that is sent by a validator to the network. The consensus message contains information about previous events that have been observed and validated, including their parents, which are specified as the parents of this event [49].

By definition, Lachesis is an asynchronous Byzantine fault tolerance (aBFT) consensus protocol. The aBFT consensus algorithm is characterized by high speed and low energy consumption, as it uses neither the energy-intensive PoW nor the round-based PoS schemes [48,50]. Lachesis utilizes the EventDAG to store and sort events with transactions and provides guaranteed and instant finality. This means that once a transaction has been confirmed, it is irreversible unless more than a third of the network validators act in a Byzantine manner. By leveraging the EventDAG, Lachesis can efficiently determine the order of transactions, thereby accelerating the consensus mechanism [50]. Fantom also uses leaderless PoS to secure the network with the staking of the native token of Fantom (FTM) and performing block validation. Unlike conventional PoS systems, this leaderless PoS does not grant validators the right to determine the validity of blocks, thereby enhancing the security of the network [48].

Consensus building in Fantom occurs in several steps: creation of events, formation of roots, election of Atropos, and arrangement of events. Each event contains transactions and is created by validators on the network. A special event in the Lachesis algorithm is the "root". A root marks the beginning of a new frame, which represents a unit of logical time within the DAG. The Atropos is the first root that was classified as a candidate and represents the final state after the consensus process. Specifically, the Atropos represents a final block for chaining to a blockchain that contains all transactions in the subgraph of the Atropos. To realize the support of smart contracts, all final Atropos blocks in Fantom are arranged into a blockchain. Using the resulting blockchain, Fantom is thus able to achieve EVM smart contract compatibility [50].

To further increase the efficiency of the system, Fantom divides the EventDAG into sub-EventDAGs, referred to as epochs. Each epoch encompasses a certain number of finalized Atropos blocks and represents a separate unit of logical time. After sealing an epoch, new events for this epoch are ignored, thereby optimizing the storage and processing of data [51].

### 3.3.2. Avalanche

Avalanche is another hybrid DLT that aims to significantly improve the scalability and speed of blockchain-based networks. It employs a consensus mechanism named Snowball. This consensus mechanism is highly scalable and allows for decentralized networks where thousands of nodes can make decisions securely and efficiently [52].

The Snowball consensus algorithm operates as follows: Firstly, certain parameters are defined that are important for the algorithm. These parameters include the number of participants ($n$), the sample size ($k$), the quorum size ($\alpha$), and the decision threshold ($\beta$). For the algorithm, two colors, blue and red, are used to represent two competing decisions. The focus is on the total number of nodes that prefer blue. As long as no decision has been made, the algorithm queries $k$ randomly selected nodes for their preferences. If $\alpha$ or more nodes give the same answer, this answer is adopted as the new preference. If this preference is the same as the old preference, a counter for consecutive successes is increased by one. However, if the preference is different, the counter is reset to one. If no answer reaches a quorum, that is, an $\alpha$ majority, the counter is reset to zero. This process is repeated until the same answer achieves a quorum $\beta$ times in a row [52,53].

Security and liveness are two important factors in a consensus protocol and can be parameterized in Avalanche. As the quorum size $\alpha$ increases, security increases while

liveness decreases. This means that the network can tolerate more faulty (Byzantine) nodes while remaining secure. In the public Avalanche network, these parameters are kept constant and fairly small. The sample size $k$ is 20 and the quorum size $\alpha$ is 14. The decision threshold $\beta$ is 20. These settings allow the Avalanche network to remain highly scalable even as the number of nodes in the network increases. Avalanche also uses a transaction-based directed acyclic graph (TxDAG) to organize vertices as transactions. The Snowball consensus mechanism is a protocol optimized for DAGs, characterized by high throughput and parallel processing [52,53].

The *Snowman* consensus protocol, on the other hand, is optimized for blockchains. Snowman also exhibits high throughput and is particularly suitable for smart contracts, as it is an implementation of Snowball that ensures a completely linear arrangement. When the Snowball consensus mechanism is initialized with a virtual machine whose state is a single unspent transaction output (UTXO) and whose transaction format only generates a single UTXO, the result is the Snowman consensus mechanism. The UTXO represents the current state, and the output UTXO represents the new state. The Avalanche network consists of several subnetworks [54].

## 4. Evaluation Setup

For the network performance evaluation, five of the most prominent DLTs, each using either a blockchain, DAG, or a hybrid approach for their internal transaction ledger, have been chosen. The following sections will provide an overview of the evaluation aspects required for an objective comparison of the DAG-based solutions.

### 4.1. Methodology

To achieve the comparison of the different DLTs, an experimental setup has been used to measure the predefined KPIs from Section 4.4. To conduct the necessary tests, private instances of the technologies have been setup, when available. The remainder has been tested using publicly available instances and networks. The utilized network parameters are described in Section 4.2 and the hardware configuration used for the tests is described in Section 4.3. Later on, these experimental results are brought into the context of a theoretical analysis of the performance, building a bridge between an experimental and theoretical analysis.

### 4.2. DLT Network Parameters

The evaluation of the performance and scalability of the different DLTs was conducted with different configurations. The blockchain technologies, namely Fabric and Ethereum, were set up on a private network. The same approach was used for IOTA as well. Only the hybrid DLTs were tested in their public test networks, allowing for comparison within a broad spectrum of implementation scenarios.

Fantom and Avalanche were evaluated using public test networks with 8 and 460 nodes, respectively. The number of nodes was chosen by the initial network operator and could not be altered. For all other tested technologies, it was possible to set up an arbitrary number of nodes/clients, allowing for a more fine-drawn comparison. An overview of the number of nodes for each DLT can be found in Table 2.

**Table 2.** DLT software version and nodes overview.

| DLT | Type | Version | Number of Nodes |
|---|---|---|---|
| Fabric | Private | 2.3 | 4, 8, 16 |
| Ethereum (Besu) | Private | 23.1.1 | 4, 8, 16 |
| IOTA | Private | Chrysalis | 2, 4, 8, 16 |
| Fantom | Public | Lachesis Test Network | 8 |
| Avalanche | Public | Snowball Test Network | 460 |

The time interval that determines when to append a new block to the given blockchain or a transaction to the given DAG, called block time, is determined individually by each technology as well. The Fantom and the Avalanche test networks have mechanisms for determining the block times dynamically, depending on the network load. This constitutes a significant difference from classical blockchain systems such as Ethereum, where the block time is usually predetermined and fixed, even in private setups. This peculiarity was taken into account in the evaluation, as it could potentially have significant effects on the observed KPIs. Fabric and IOTA also use predetermined intervals for their block and transaction times, respectively.

To extend the performance comparison even further, we also conducted evaluations of different Ethereum test network configurations. As mentioned in Section 3, the Hyperledger Besu test environment allows a high degree of control over the test network. Therefore, two different consensus algorithms could be compared: Istanbul Byzantine fault tolerance (IBFT) and Quorum Byzantine fault tolerance (QBFT).

The number of nodes/clients in the private test networks was chosen to be 4, 8, and 16, allowing us to extrapolate the scaling effects to node values beyond the tested scenarios. Furthermore, this enables direct comparability to previous performance evaluations [10,11,55,56]. For each individual test network, the block times were set to one second. The IOTA test network in its current state requires a centralized coordinator that issues milestones, thereby finalizing all transactions in the network. In our test scenarios, the coordinator issues a new milestone every minute, following the public network implementation of IOTA.

### 4.3. Hardware Configuration

The DLTs Fantom and Avalanche are evaluated in their public variants; thus, precise control over the hardware is not possible. Thus, this evaluation will focus on the real-world performance numbers rather than evaluating the performance on a specific hardware platform, such as Intel or ARM. However, to mimic the heterogeneous behavior of public DLT networks, the following 64-bit hardware scenarios (cf. Table 3) were used to evaluate the different DLTs. Table 2 provides an overview of the given software versions used during the evaluation.

**Table 3.** Hardware overview.

| DLT | Processor | RAM | | OS |
| --- | --- | --- | --- | --- |
| Fabric | 24C/24T 3.8 GHz Intel | 128 GiB 2133 MT/s | DDR4 | Ubuntu 16.04 LTS |
| Ethereum | 12C/24T 3.8 GHz AMD | 31.8 GiB 3200 MT/s | DDR4 | Arch Linux 6.3.1-arch2-1 |
| IOTA | 14C/20T 2.3 GHz Intel | 14.9 GiB 5600 MT/s | DDR5 | Microsoft Windows 11 Home, 10.0.22621 (Run on WSL 2) |

All the evaluated technologies are under current development. IOTA, in particular, is currently transitioning from a coordinator-reliant network setup to a fully decentralized architecture. Thus, the performance results might vary depending on the time of reading.

To set up the different networks, selected tools were used to facilitate rapid setup and configuration. IOTA provides a project called "one-click-tangle" containing utilities to set up a private network. This project has been used in order to set up the private IOTA network for the tests of this paper. The full project is available on GitHub through [57]. The nodes have been set up with a PoW score of 4000, which is used by the main network of IOTA as well [58]. Fabric was tested and evaluated using a custom-written generator framework available on GitHub [59]. The Besu (Ethereum) network has been set up from scratch for each test scenario using the provided software development kit (SDK).

### 4.4. Key Performance Indicators

To quantify the performance of the different DLTs, the most common network performance indicators *TPS* and *latency* were chosen as KPIs. Previously conducted performance evaluations also used the same KPIs, thus allowing the comparison of these results to the concrete numbers presented in this article. Since each technology has its own feature set and configurable parameters, other, more specific KPIs are also considerable. However, for the sake of objectivity and comparability, the common KPIs TPS and latency were chosen. It shall be noted that their individual definitions differ slightly between the different technologies. Moreover, in the case of IOTA, the concrete hardware characteristics were measured to obtain an overall impression of the real-world implications and potential resource constraints. Therefore, we define all of these terms objectively as follows:

- Transactions per second (TPS).

The absolute number of transactions that the given network can validate, finalize, and append to the given ledger within one second. A transaction, in this particular case, refers to the transition of one ledger state to a new state.

- Latency.

The absolute time required for one transaction to be finalized and written to the ledger. In other words, the time it takes for the transferred data to be available to and verifiable by any other network participant. Our definition of latency is equivalent to the term time to finality (TTF), which is the time required to finalize the transaction and make it available to each network participant.

- (Optional) hardware characteristics.

The hardware characteristics are evaluated on a Raspberry Pi 3 using containerized nodes. The key metrics are measured in terms of central processing unit (CPU) utilization (%), random access memory (RAM) usage (MiB), and power consumption (watts) for IOTA only. An explicit listing of the measured values will not be provided by this paper, as these characteristics do not mark the focus of this paper and were only evaluated to obtain an overall trend of resource consumption in this particular test scenario. The paper focuses on the evaluation of TPS and latency while also providing a first impression of resource consumption.

### 5. Performance Evaluation

Objectively evaluating the selected DLTs has proven to be a significant challenge since the major KPI concepts, described in Section 4.4, have to be adjusted slightly to fit the intricacies of each technology. First, a clear distinction between private and public DLTs has been created. In the private category, IOTA, Fabric, and Ethereum are compared with each other, and in the public category, Fantom and Avalanche are compared. All performance tests have been repeated and validated with a total of $n = 1000$ runs. Each transaction within the IOTA test network used a one-byte payload, thus requiring 4000 hashing operations on average for the PoW.

### 5.1. Private DLT Performance Comparison

When comparing the private DLT solutions with each other, IOTA poses some challenges to the objectivity of the comparison, whereas Hyperledger Fabric and Ethereum Besu can be compared directly in terms of TPS or latency. While Fabric and Ethereum organize their internal ledger as a blockchain, IOTA utilizes a DAG. This distinction is a major point to consider since the main bottleneck of blockchain TPS performance is the limit of one block that can be appended at a given point in time. Using a DAG, an arbitrary number of transactions can be appended, in theory. Thus, the aforementioned bottleneck is eliminated entirely with this approach, and the comparison presented here indicates the benefits of using a DAG instead of a blockchain.

In consequence, calculating the TPS limit of IOTA is challenging since no hard limit exists. Rather, the TPS will scale with the number of nodes issuing transactions in the network. The current implementation of IOTA uses a centralized coordinator that finalizes the transactions after a previously configured amount of time. Therefore, the real TTF is currently determined by this parameter. However, since one of the goals of the IOTA Foundation is to replace the coordinator with a decentralized approach, the following evaluation results will only focus on the processing times for each transaction and will intentionally leave out this static configuration.

Figure 3 provides an overview of the TPS measurement results for each technology. Each technology was evaluated with three different numbers of nodes in the respective test networks. First, the IOTA results show that there is indeed no noticeable scaling effect among the test cases. In an ideally load-balanced network, each measurement can be multiplied by the number of unique nodes, thus resulting in the real TPS that the network can process. Furthermore, the results also allow the differentiation of the Ethereum TPS in terms of the consensus protocol (IBFT and QBFT). However, no significant performance impact can be determined for the given network setups. When comparing the Ethereum TPS with the respective HLF TPS results, Fabric's mean TPS are over 1.7 times higher than those of Ethereum on average, thereby confirming previously found results [9,21].

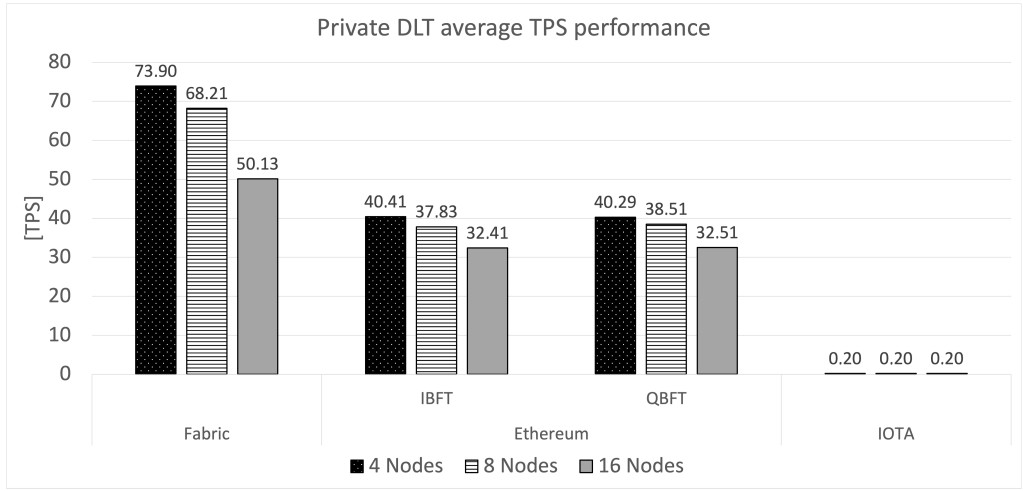

**Figure 3.** Private DLT TPS performance.

In a fair comparison, a load-balanced IOTA network consisting of at least 370 nodes that each send the same amount of transactions is capable of achieving the same TPS as Fabric. Likewise, with more than 202 nodes, IOTA's performance surpasses the maximum TPS of Ethereum. One remarkable aspect is that the conducted tests used a single client connected to a single node of IOTA. The measured time to calculate the TPS is the time of the client blocking for processing the transaction, including the PoW calculation. Hence, IOTA's measured TPS is the TPS of a single client. Multiple clients on a single node of IOTA should therefore also further increase the overall TPS of the network. In addition to that, IOTA's PoW correlates with the length of the message and directly influences a client's TPS. Decreasing the difficulty would result in greater throughput. Additionally, it shall be noted that, generally, increasing the number of nodes in a blockchain-based network will result in diminished TPS numbers (cf. Figure 3). Therefore, the scaling behavior of IOTA is significantly better than Fabric's or Ethereum's.

When comparing the latency times of the different private DLTs to each other, IOTA's latency times also have a distinct meaning to them. Since there is no artificial block limitation or race between the nodes to append a transaction, the only latency that can be measured is the time it takes to write a valid transaction to the tangle. This time is directly proportional to the required number of hashing operations, determined by the PoW score.

In our test case, we used a fixed PoW score of 4000, which is used by the official public IOTA network as well. As Figure 4 shows, no scaling effects occur due to the internal structure of IOTA.

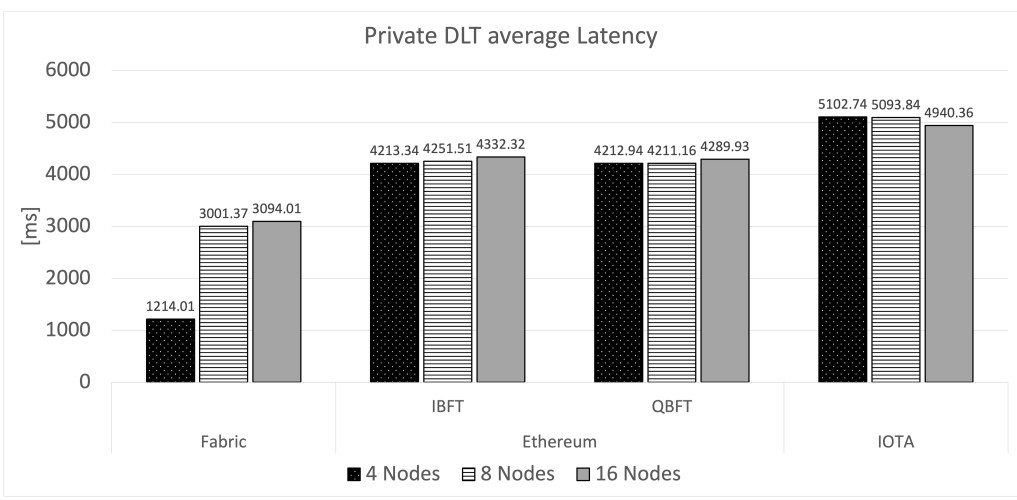

**Figure 4.** Private DLT latencies.

The other private DLTs solutions do show a negative scaling trend and increased latency times when including more nodes within the network. Here, the latency times are also determined by the time it takes a node to craft a valid transaction and submit it to the network. In these cases, the latter aspect is of notable importance since the blockchain creates a latency bottleneck due to its linear structure.

The overall resource consumption of IOTA appears to be low in comparison to other DLTs. Nodes tend to use around 150 MiB of memory and only a fraction of the CPU. The normal energy consumption on a Raspberry Pi 3 of an IOTA node is not significantly higher than the baseline consumption of the device. However, using the PoW of IOTA increases the computational resource consumption drastically by design, as well as the correlated energy consumption.

*5.2. Public DLT Performance Comparison*

In order to show a broad spectrum of DLT performance numbers, hybrid public DLT solutions were also evaluated in terms of latency and TPS. Both Fantom and Avalanche use a hybrid DLT approach, which stores the new transactions in a main blockchain but uses a DAG for the consensus operation. This generally leads to increased consensus performance but keeps the simplicity and compatibility of existing blockchain ecosystems, most notably the EVM for the execution of smart contracts.

In our evaluation results, visualized in Figure 5, the Avalanche Snowball network is significantly slower than the Fantom Lachesis test network in terms of latency. It should be noted that the Fantom network only contains a fraction of the nodes contained in the Avalanche network; thus, no fair direct comparison can be made between the networks. Nevertheless, the TPS numbers are also higher than for private DLTs. Since both Fantom and Avalanche utilize faster DAG-based consensus operations, higher throughput rates can be achieved, thereby increasing the TPS of the network. Furthermore, neither technology relies on PoW consensus operations to create a valid transaction but rather uses PBFT protocol, omitting the potentially long hashing times.

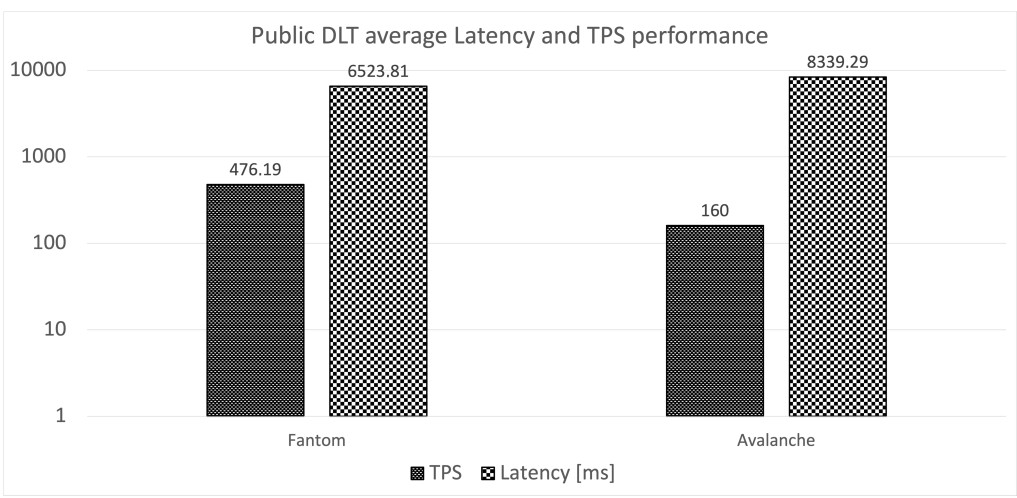

**Figure 5.** Public DLT average TPS and latency times.

*5.3. Discussion*

The performance evaluation of all DLT variants indicated a positive performance trend when the given technology uses a DAG, at least for the consensus operation. The commonly used blockchain data structure, which in itself is also a special DAG with a fixed predecessor count of one (one-dimensional DAG), can be a bottleneck when faced with a high number of TPS. This is due to (1) the fixed number of transactions that can be written to a given block and (2) the limit of only one block that can be appended at once.

By increasing the allowed number of predecessors for a block (or a transaction, in the case of IOTA), the second bottleneck is eliminated. By allowing multiple validation and appending operations in parallel, significantly higher throughput performance can be achieved. This is exactly the case with Fantom and Avalanche, which still rely on a fixed transaction limit per block but allow more predecessors in their consensus DAG.

IOTA fundamentally omits the number of transactions per block by eliminating the concept of a block entirely in favor of a pure transactional DAG. Thus, the IOTA tangle contains only single transactions being arranged in a decentralized DAG structure. This structure allows maximum parallelization and, thus, a theoretically arbitrary scaling potential. Other blockchain-based DLTs will experience diminished performance with an increasing number of network participants, whereas DAG-based solutions will be able to compensate for the increased load and scale well.

In its current development state, the IOTA network is still reliant on a central coordinator, which reduces the decentralization of the network. Again, due to this mechanism, transactions are only valid after they have been referenced in a milestone by the coordinator. Since these milestones are created in a fixed time interval, the real TTF is simply the mentioned fixed milestone creation interval. In future versions, this concept will be replaced in favor of a pure decentralized approach. One notable aspect is that the conducted tests for IOTA measured the performance of a single client and not the network in general.

The overall resource consumption of IOTA is lower compared to traditional DLT solutions. A natural exception is the fact that IOTA uses a classical PoW, as already stated above. Since the PoW is designed to intentionally consume computational resources, resource consumption is tied to the use of the PoW algorithm. To further influence the use of computational resources, the difficulty of the PoW could be decreased on the network or the calculation could be outsourced to the IOTA node if the configuration of the node allows this feature. Overall, IOTA can be seen as resource efficient for use with devices that have limited resource availability. Overall, this paper agrees with the findings of [4,19] for the practically lower throughput. In comparison, this paper identifies the configuration of the PoW as the main influence on network performance.

Generally, deciding on a given technology or platform is a non-trivial task. Most use cases that can benefit from the introduction of DLTs are retrofitting this technology,

thereby replacing already existing database systems, e.g., relational or time series databases. However, these technologies are designed to cope with multiple thousands of transactions per second, while common DLT platforms fall significantly behind these performance numbers. Therefore, if a developer aims to implement, e.g., one of the aforementioned DLTs in their system, considering the necessary amount of transactions per second is a necessary task. Also, since our findings support the assumption that DAG-based platforms scale their performance positively and linearly with an increasing amount of network participants, including more network participants may be beneficial or even necessary in some scenarios. The case of blockchain-based systems is more simple in this regard since these do reach a maximum throughput threshold, e.g., at 30 TPS on average in the case of Ethereum. Thus, a developer will need to determine the expected number of transactions the DLT system needs to handle as well as the number of added network participants.

## 6. Application Scenarios

This section presents two real-world use case scenarios that can benefit from introducing DLTs for data storage. Both scenarios share the common requirement to store data that must not be modified after initial creation. Furthermore, third parties shall only be able to see and/or validate the data gathered in both use cases. The first scenario is settled in the domain of logistics, where information about the origin and transport chain of a given product shall be made transparent to the consumers. The second example describes a smart energy grid scenario involving energy "prosumers", such as battery storage or electric vehicles, which are controlled remotely to optimize the usage of volatile energy sources.

### 6.1. Supply Chain Traceability

DLTs present opportunities to enhance transparency and traceability of products within (food) supply chains. By utilizing DLTs, product information can be securely and immutably stored, thereby facilitating the entire process of traceability and verification [60]. DLTs provide a decentralized infrastructure, wherein all transactions and data across the supply chain can be recorded. Every actor within the supply chain, such as seed sellers, farmers, producers, wholesalers, retailers, regulatory authorities, and consumers, can have access to supply chain data and verify the information [61]. The stakeholders are illustrated here using the example of a food supply chain, as shown in Figure 6, but the concept can be adapted to any means of supply chain. The integrity of the data is ensured through the cryptographic design of the DLT. This offers several advantages for the management of (food) supply chains:

- Transparency and traceability: With DLTs, all product-related information, from seed seller to end-consumer, can be logged. This allows for complete transparency and traceability of the product throughout the supply chain, thereby boosting consumer and regulatory trust.
- Data integrity: The decentralized nature of DLTs and the application of cryptographic techniques ensure that data stored in the blockchain cannot be manipulated or altered, thus guaranteeing data integrity and protection against fraud.
- Automation and efficiency: By leveraging smart contracts, certain processes within the supply chain can be automated. This can lead to significant efficiency gains and a reduction in human errors.
- Interoperability: DLTs can be integrated with the Electronic Product Code Information Services (EPCIS) [62] standard to enable better interoperability of data throughout the entire supply chain. This facilitates data collection, analysis, and utilization and aids in meeting compliance requirements.

However, the problem of the scalability of DLTs, or rather the lack of it, arises. Simply using a private DLT with a limited number of users is unfeasible in this scenario as the group of end consumers should be as unlimited/open as possible. One approach to tackling the problem is the reduction of the amount of data that needs to be stored on-chain, i.e., in the DLT. A possible solution can be the use of InterPlanetary File System (IPFS) as a storage

system. As each data item stored in an IPFS is identified by a content identifier (CID), which also ensures immutability [63], the CID can be stored in the DLT instead. The IPFS is a peer-to-peer storage system, where each chunk of data is addressed by its CID, which itself is the SHA-256 hash value of the data. This addressing scheme protects the data from modifications by a malicious party.

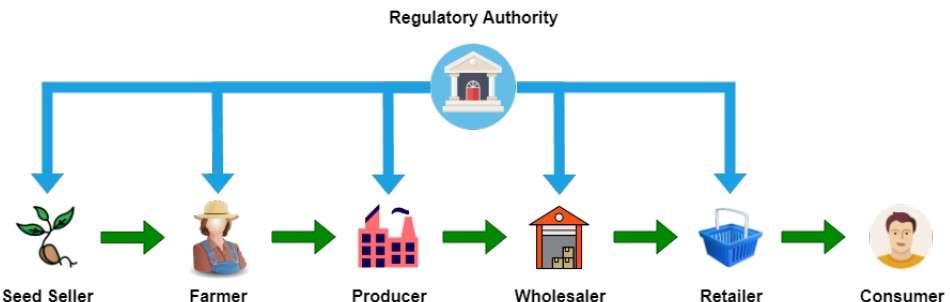

**Figure 6.** Actors and flow of goods in a food supply chain.

### 6.2. Trusted Smart Grid Communication

Energy networks, such as the German electric grid, are currently undergoing a decentralization process. A framework for smart metering has been standardized in order to automate the metering and introduce the ability to control decentralized energy producers and consumers. This system consists of a smart meter gateway (SMGW) placed in a customer's building. This device is responsible for managing the communication between the customer's devices, such as local meters, producers, and consumers, the network provider, and other participants in the smart grid [64]. These local producers and consumers are called controllable local systems (CLS). These CLS are located behind the SMGW and use it for different communication tasks across three different networks. The local meterological network (LMN) contains the local energy meters (also named smart meters), the home area network (HAN) connects, among others, the CLS, and the wide area network (WAN) manages the connection to the outside world, e.g., the Internet. The latter includes a gateway administrator (GWA) managing the SMGW and other external organizations. These can receive the measured energy production and consumption values and can be authorized to control the CLS [65]. The control of such a CLS device is regulated in the German Federal Energy Economy Act in §14a. Already, the overall system must log different data, for example, errors and updates on the SMGW. However, current regulations do not specify adequate logging of switching operations when controlling a CLS [65]. The control of these systems might be relevant for billing purposes, and thus the logging of switching operations bears the risk of being manipulated by a dishonest party. Therefore, neither the customer nor the external organization shall be responsible for the logging because both of them might try to manipulate the logged data in order to obtain financial benefits. A possible solution for this might be logging within a DLT.

In such a setup, the nodes of the DLT should be operated within the HAN, i.e., logically behind the SMGW, and can be operated by an already existing device, like the SMGW itself or a CLS device. The general idea is that the client software handles the collected log messages from the CLS and stores them in a local database. Any changes to this local database will be committed to the DLT. This architecture is comparable to the world state paradigm of Hyperledger Fabric.

The log messages are gathered and handled locally. Thus, the local database can be reverted to any state in history by using the data stored on the ledger. Using this mechanism, the system is not bound to a specific DLT implementation, making it possible to be exchanged by any other compatible DLT, e.g., exchanging a blockchain platform for a DAG-based DLT.

Since the smart grid is considered to be a dynamic network with a high degree of joining and leaving parties, scaling the network is a considerable factor in choosing the

appropriate DLT. Since IOTA has been shown to be significantly more capable of handling a larger number of clients while also allowing reliable throughput performance, it has been chosen as a pure DAG-based DLT for use in the anticipated logging system proof of concept (PoC) in the first implementation phase. To prove the ability to exchange the DLT backend, Hyperledger Fabric has been used in the second phase of the implementation. The PoC demonstrator showed that, due to the use of the world-state paradigm, the DLT implementation exchange was directly possible.

One remaining problem is addressing the authenticity of a single CLS device to prove that a certain log message was clearly issued by a single identifiable device. Therefore, a PKI has been included within the network, ensuring authenticity by issuing and distributing cryptographic certificates. Every log message-issuing device is identified by a public key, and the signature is created with the private key. A certificate for the relation between a public key and a device is issued by the aforementioned PKI and also stored on the DLT. Other participants in the network can validate every single log message using the PKI-provided certificates. The revocation of a single certificate is also handled by the PKI and stored on the DLT.

## 7. Conclusions

This article presents a comprehensive study of the concrete, real-world performances of modern DLTs. In particular, three different categories, blockchain, DAG, and hybrid DLTs, have been evaluated with regard to their throughput and latency. For each category, the most popular or well-supported implementations were chosen to allow a high degree of applicability of the article.

The findings suggest that the singular node and client performance of DAG-based DLTs, particularly IOTA's implementation, is significantly lower than both blockchain and hybrid DLTs in terms of throughput and latency. Nevertheless, it excels when scaling the network to multiple nodes working in parallel. When compared directly to blockchain-based technologies like Hyperledger Fabric or Ethereum, which do experience with decreased performance when scaled to multiple nodes within the network, IOTA does not behave in the same way. On the contrary, IOTA even gains throughput performance and remains at the same latency level, no matter the node scaling. Thus, for smaller networking scenarios with a fixed amount of nodes within the network, blockchain or hybrid DLTs provide the most reliable throughput and latency, whereas, in dynamically changing scenarios, DAG-based DLTs should be used to cope with higher flexibility and scalability demands. Hybrid DLTs provide an intermediary solution, fixing both the scalability problem of blockchain systems and the diminished throughput performance of DAG-based DLTs. These technologies have been evaluated in their public, generally slower variants. But even though their implementation relies on the whole internet, they have proven to be viable candidates to implement smart contracts on or deploy custom nodes. Due to their DAG-based consensus, they are generally more capable of handling multiple hundreds or thousands of transactions per second, which has previously not been possible with traditional blockchain systems.

As shown in the real-world use case examples, even existing scenarios can benefit from the introduction of DAG-based DLTs, e.g., to host trusted data or provide traceability for legal product provenance. Future work may focus on setting up all-private DLTs in the above-mentioned categories to provide a private DLT network comparison. Furthermore, since IOTA is currently transitioning to a fully decentralized version, also allowing smart contract integration, comparing Fabric's, Ethereum's, and IOTA's smart contract performance to each other could result in a comprehensive performance overview for interested developers.

In the near future, more DAG technologies will need to emerge and evolve in order to increase both research efforts and industrial appreciation of this technological paradigm. The improved scalability and generally higher transaction throughput form the basis for a highly competitive technology for blockchain platforms. By adding smart contract

functionality, fully functional DAG platforms can truly outperform common blockchain systems in terms of both scalability and performance.

**Author Contributions:** Conceptualization, M.F. and J.D.; Data curation, F.H. and F.K.; Formal analysis, F.H., F.K., M.F. and J.D.; Funding acquisition, R.T.; Investigation, F.H. and F.K.; Methodology, M.F. and J.D.; Project administration, R.T.; Resources, R.T., M.F. and J.D.; Software, F.H. and F.K.; Validation, J.D.; Visualization, F.H., F.K. and J.D.; Writing—original draft, F.H. and F.K.; Writing—review and editing, M.F. and J.D. All authors have read and agreed to the published version of the manuscript.

**Funding:** This work is funded in part by the Federal Ministry of Education and Research Germany under grant numbers 16KIS1701 and 16KIS1540, and in part by the Federal Ministry of Economic Affairs and Climate Action Germany under grant number 03EI6083.

**Data Availability Statement:** All datasets can be found on Github [66].

**Conflicts of Interest:** The authors declare no conflict of interest.

## Abbreviations

The following abbreviations are used in this manuscript:

| | |
|---|---|
| **aBFT** | asynchronous Byzantine fault tolerance |
| **CA** | certification authority |
| **CID** | content identifier |
| **CLS** | controllable local systems |
| **CPU** | central processing unit |
| **DAG** | directed acyclic graph |
| **DLT** | distributed ledger technology |
| **EPCIS** | Electronic Product Code Information Services |
| **EventDAG** | event-based directed acyclic graph |
| **EVM** | Ethereum virtual machine |
| **GWA** | gateway administrator |
| **HAN** | home area network |
| **HLF** | Hyperledger Fabric |
| **IBFT** | Istanbul Byzantine fault tolerance |
| **IPFS** | InterPlanetary File System |
| **IoT** | Internet of Things |
| **KPI** | key performance indicator |
| **LMN** | local meterological network |
| **M2M** | machine-to-machine |
| **PoA** | proof-of-authority |
| **PoC** | proof of concept |
| **PoS** | proof-of-stake |
| **PoW** | proof-of-work |
| **PBFT** | practical Byzantine fault tolerance |
| **PKI** | public–key infrastructure |
| **QBFT** | Quorum Byzantine fault tolerance |
| **RAM** | random access memory |
| **SDK** | software development kit |
| **SMGW** | smart meter gateway |

| TxDAG | transaction-based directed acyclic graph |
|---|---|
| **TPS** | transactions per second |
| **TTF** | time to finality |
| **UTXO** | unspent transaction output |
| **WAN** | wide area network |

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
