# Peer review of "Performance Comparison of Directed Acyclic Graph-Based Distributed Ledgers and Blockchain Platforms"

_computers, doi:10.3390/computers12120257_

Round 1

Reviewer 1 Report

Comments and Suggestions for Authors

The abstract provides a clear overview of the paper's topic, which is comparing directed acyclic graph (DAG) platforms with traditional blockchain systems in terms of performance. However, it could benefit from a brief statement about the significance of the research and its potential impact on the field of distributed ledger technology.

The introduction effectively sets the stage by outlining the growth of blockchain ecosystems and the desire to scale these systems for high transaction throughput. However, it could provide a bit more context on the challenges faced by traditional blockchains in achieving high throughput and why DAGs are seen as a potential solution.

The paper's research objective, i.e., comparing the performance of DAG platforms to traditional blockchains in terms of transaction throughput and network latency, is clearly stated. It would be beneficial to explicitly mention any hypotheses or research questions the study aims to address.

The methodology section is crucial but not outlined in the abstract. Including a brief mention of the research methods used in the abstract would give potential readers an idea of how the comparison was conducted.

I recommend the following issues:

-          clarify the main research question and objectives to guide the reader through the paper's focus.

-          Provide a bit more context in the introduction about the specific challenges faced by traditional blockchains in achieving high transaction throughput and why directed acyclic graphs (DAGs) are being explored as a potential solution. This will help readers understand the background of the research more clearly.

-          Explicitly state any hypotheses or research questions that the study aims to address. This will make the research objectives more specific and clear.

-          Include a brief mention of the research methods used in the abstract. This will give potential readers an idea of how the comparison between DAG platforms and traditional blockchains was conducted.

-          need to improve the list of references by providing sufficient background and including these relevant references  -

·       Amina Y. Alsallut, Ruba A. Salamah, Aiman A. Abusamra, "Exploratory Study on Hyperledger Fabric Framework: Food Supply Chain as a Case Study", International Journal of Engineering and Manufacturing (IJEM), Vol.13, No.4, pp. 11-19, 2023. DOI:10.5815/ijem.2023.04.02

·       Oishi Chowdhury, Md Al Samiul Amin Rishat, Md. Al-Amin, Md. Hanif Bin Azam, " The Decentralized Shariah-Based Banking System in Bangladesh Using Block-chain Technology", International Journal of Information Engineering and Electronic Business(IJIEEB), Vol.15, No.3, pp. 12-28, 2023. DOI:10.5815/ijieeb.2023.03.02

-          The platforms that were compared or at least indicate the number of platforms. This can help readers understand the breadth of the comparative analysis.

-          Include a brief statement about potential future directions for fully DAG-based platforms. This can help readers understand the potential for development and improvement in this area.

-          Expand on the practical implications of the findings, particularly in terms of decision-making for developers. Explain why the higher transaction throughput of DAG-based solutions matters and how it could impact their projects or systems.

-          Briefly mention what the two real-world application scenarios benefiting from DAG-based technologies are in the abstract. This provides more context and demonstrates the practical relevance of the research.

-          While the abstract is well-written, consider making it slightly more concise without sacrificing important information. Aim for clarity and impact in a brief format.

Comments on the Quality of English Language

Minor editing of English language required

Author Response

Dear Reviewer,
we would like to thank you for reviewing our Paper "Performance Comparison of Directed-Acyclic-Graph-based Distributed Ledgers and Blockchain Platforms". We have read your comments and really appreciate your effort.

Please allow us to comment on your suggestions:
- clarify the main research question and objectives to guide the reader through the paper's focus.
> We totally agree here! We added a paragraph in the introduction stating the main research question and focus.

- Provide a bit more context in the introduction about the specific challenges faced by traditional blockchains in achieving high transaction throughput and why directed acyclic graphs (DAGs) are being explored as a potential solution. This will help readers understand the background of the research more clearly.
> Good point. We added a section in the introduction, addressing this and rewrote some parts to better explain the scalability challenges. 

- Explicitly state any hypotheses or research questions that the study aims to address. This will make the research objectives more specific and clear.
> This refers to point one, thus fixed.

- Include a brief mention of the research methods used in the abstract. This will give potential readers an idea of how the comparison between DAG platforms and traditional blockchains was conducted.
> Done.

- need to improve the list of references by providing sufficient background and including these relevant references  -

    · Amina Y. Alsallut, Ruba A. Salamah, Aiman A. Abusamra, "Exploratory Study on Hyperledger Fabric Framework: Food Supply Chain as a Case Study", International Journal of Engineering and Manufacturing (IJEM), Vol.13, No.4, pp. 11-19, 2023. DOI:10.5815/ijem.2023.04.02

    · Oishi Chowdhury, Md Al Samiul Amin Rishat, Md. Al-Amin, Md. Hanif Bin Azam, " The Decentralized Shariah-Based Banking System in Bangladesh Using Block-chain Technology", International Journal of Information Engineering and Electronic Business(IJIEEB), Vol.15, No.3, pp. 12-28, 2023. DOI:10.5815/ijieeb.2023.03.02
> After discussing this internally, we chose to include your first reference. We refrained from including the second reference due to neutrality reasons in terms of religion and culture.

- The platforms that were compared or at least indicate the number of platforms. This can help readers understand the breadth of the comparative analysis.
> We added the concrete number "five technologies" when speaking about "broad spectrum of technologies"

- Include a brief statement about potential future directions for fully DAG-based platforms. This can help readers understand the potential for development and improvement in this area.
> We added a section in the conclusion addressing the future directions of DAG-based platforms, in case that they become fully evolved.

- Expand on the practical implications of the findings, particularly in terms of decision-making for developers. Explain why the higher transaction throughput of DAG-based solutions matters and how it could impact their projects or systems.
> Good point again! We included an additional paragraph in the discussion section 5.3 addressing these points.

- Briefly mention what the two real-world application scenarios benefiting from DAG-based technologies are in the abstract. This provides more context and demonstrates the practical relevance of the research.
> Done. 

- While the abstract is well-written, consider making it slightly more concise without sacrificing important information. Aim for clarity and impact in a brief format.
> We rewrote the abstract and hope that this meets your expectations.

With this, we hope to have addressed all your issues with this paper. Thanks again for reviewing and sharing your thoughts with us. We really appreciate the effort.

Best regards,

the Authors

Reviewer 2 Report

Comments and Suggestions for Authors

This paper compares the performance of common DLT platforms using the most commonly used metrics. The evaluation suggests that IOTA (a DAG-based platform) has significantly lower throughput and latency compared to other blockchain platforms. However, it excels when scaling the network to multiple nodes working in parallel. The results are clear and convincing. However, the novelty and the obvious connections between DAG throughput, latency, and scalability are known advancements, which limit the significance of this work. I appreciate the quality of the work and the good presentation of this paper.

I suggest the author to consider adding more novelty to this work and show insights which were not commonly recongized. Also, blockchain with sharding mechanism should also be considered, as they also scale up well. 

Examples of sharding work: 

D. Yu, H. Xu, L. Zhang, B. Cao and M. A. Imran, "Security Analysis of Sharding in the Blockchain System," 2021 IEEE 32nd Annual International Symposium on Personal, Indoor and Mobile Radio Communications (PIMRC), Helsinki, Finland, 2021, pp. 1030-1035, doi: 10.1109/PIMRC50174.2021.9569351.

Xu, H., Sun, Y., Zhang, X., Liu, E., & Chih-Lin I. (2023). When Web 3.0 Meets Reality: A Hyperdimensional Fractal Polytope P2P Ecosystems. ArXiv, abs/2308.06829.

Comments on the Quality of English Language

The english is fine. Minor grammar issues can be fixed in future revision and proof. 

Author Response

Dear Reviewer,
we would like to thank you for reviewing our Paper "Performance Comparison of Directed-Acyclic-Graph-based Distributed Ledgers and Blockchain Platforms". We have read your comments and really appreciate your effort.

Please allow us to comment on your suggestions:

"I suggest the author to consider adding more novelty to this work and show insights which were not commonly recongized. Also, blockchain with sharding mechanism should also be considered, as they also scale up well. "

> Indeed, sharding is a significant option to improve scalability of the network. We added the mentioned paper in the section addressing sharding in Ethereum 2.0. We hope, this meets your expectations in this regard.

Again, thank you very much for the review. We hope that all of your issues with the paper have been addressed.

Best regards,

the Authors

Reviewer 3 Report

Comments and Suggestions for Authors

Summary/Contribution: The paper reviews directed acyclic graph (DAG) platforms and evaluates their performance indicators in terms of transaction throughput and network latency, finding that DAG-based solutions offer significantly higher transaction throughput compared to blockchain-based platforms.

Comments/Suggestions:

  • Provide more details about the specific DAG platforms and blockchain platforms that were evaluated in the study.
  • Include a discussion on the security aspects of DAG-based solutions compared to blockchain-based platforms.
  • Consider including a section on the limitations and challenges of implementing DAG-based solutions in real-world scenarios.
  • Provide a comparison of the resource requirements (e.g., storage, computational power) between DAG-based and blockchain-based platforms.
  • Include a discussion on the potential impact of DAG-based solutions on existing blockchain ecosystems and their interoperability.
  • Consider including a section on the energy efficiency of DAG-based solutions compared to blockchain-based platforms.
  • Provide more details on the methodology used for performance measurements and how the test networks were set up for each platform.
  • Include a discussion on the scalability of DAG-based solutions and their ability to handle increasing numbers of network participants. 
  • Formal methods can be used to verify the correctness of smart contracts and blockchain codes, which can help to prevent costly errors and security breaches. Therefore, it is important to discuss the use of formal methods in your paper. 
  • For this purpose, the authors may include the following interesting references (and others):

    a. https://ieeexplore.ieee.org/document/9970534

    b. https://ieeexplore.ieee.org/document/8328737
Comments on the Quality of English Language
  • Proofread the paper thoroughly to eliminate any spelling or typographical errors.

Author Response

Dear Reviewer,
we would like to thank you for reviewing our Paper “Performance Comparison of Directed-Acyclic-Graph-based Distributed Ledgers and Blockchain Platforms”. We have read your comments and really appreciate your effort!

Please allow us to comment on your suggestions:
“Provide more details about the specific DAG platforms and blockchain platforms that were evaluated in the study.”
> We added a comprehensive table listing all evaluated technologies and their respective DLT category

“Include a discussion on the security aspects of DAG-based solutions compared to blockchain-based platforms.”
> Since this paper aims to focus on the pure performance of these technologies, discussing the particular security of the individual technologies would be out of scope of this paper. However, future work may focus on examining this point in particular. We hope you can understand.

“Consider including a section on the limitations and challenges of implementing DAG-based solutions in real-world scenarios.”
> In our discussion section 5.3 we briefly describe the current challenges of IOTA not being ready for production yet since it lacks the full decentralization as well as any smart contract capability. Future work may come up with a comprehensive overview of all challenges of DAGs, or IOTA in particular

“Provide a comparison of the resource requirements (e.g., storage, computational power) between DAG-based and blockchain-based platforms.” 
> We updated our results to include a section addressing the hardware characteristics. However, since the evaluation platforms are so significantly different by design, measuring the hardware resource requirements provides non-comparable results. Therefore, we decided on providing a general idea of the resource consumption of IOTA in particular since this DAG technology is of primary interest in this paper. We hope that this decision meets your expectations.

“Include a discussion on the potential impact of DAG-based solutions on existing blockchain ecosystems and their interoperability.”
> We added a paragraph in our discussion section addressing this aspect.

“Consider including a section on the energy efficiency of DAG-based solutions compared to blockchain-based platforms.”
> As mentioned previously, since DAG platforms are not as mature as Blockchain platforms as of writing this comment, comparing the energy efficiencies of both paradigms would yield no real-world applicable results.

“Provide more details on the methodology used for performance measurements and how the test networks were set up for each platform.”
> Honestly, we cannot see any further aspects missing in this regard since section 4 "Evaluation Setup" already describes all applied software and hardware configurations in detail. All networks were set up by following the official documentation of these technologies, if not otherwise stated.

“Include a discussion on the scalability of DAG-based solutions and their ability to handle increasing numbers of network participants.“
> Section 5.3 discusses this point.

“Formal methods can be used to verify the correctness of smart contracts and blockchain codes, which can help to prevent costly errors and security breaches. Therefore, it is important to discuss the use of formal methods in your paper.”
> As mentioned previously, IOTA does not offer any Smart Contract functionality. Thus, formal methods to verify smart contracts is entirely out of scope of this paper.

Again, thank you for your detailed review and your suggestions. We hope that our changes and comments are understandable and meet your expectations. 

Best regards,
the Authors

Round 2

Reviewer 1 Report

Comments and Suggestions for Authors

Accept in present form

Comments on the Quality of English Language

 Minor editing of English language required

Author Response

Dear Reviewer,
thank you very much for your positive feedback on our paper "Performance Comparison of Directed-Acyclic-Graph-based Distributed Ledgers and Blockchain Platforms". We really appreciate your effort of your continuous review of the paper.

Best regards,
the authors.

Reviewer 2 Report

Comments and Suggestions for Authors

The expected novelty does not present in the current revision which limits its potential of contributions to the research community. 

Comments on the Quality of English Language

Minor errors detected. 

Author Response

Dear Reviewer,
thank you very much for your positive feedback on our paper "Performance Comparison of Directed-Acyclic-Graph-based Distributed Ledgers and Blockchain Platforms". We really appreciate your effort of your continuous review of the paper.

We outlined the novelty by going into more detail of the differences of the related work. We hope that this fulfills you needs.

Best regards,
the authors.

Reviewer 3 Report

Comments and Suggestions for Authors

The authors considered my comments and suggestions. Good luck.

Comments on the Quality of English Language

May be improved.

Author Response

(The authors gave the same response as above.)
